# Gather-Excite: Exploiting Feature Context in Convolutional Neural Networks

**Jie Hu**[*]
Momenta
hujie@momenta.ai

**Li Shen**[*]
Visual Geometry Group
University of Oxford
lishen@robots.ox.ac.uk

**Samuel Albanie**[*]
Visual Geometry Group
University of Oxford
albanie@robots.ox.ac.uk

**Gang Sun**
Momenta
sungang@momenta.ai

**Andrea Vedaldi**
Visual Geometry Group
University of Oxford
vedaldi@robots.ox.ac.uk

## Abstract

While the use of bottom-up local operators in convolutional neural networks (CNNs) matches well some of the statistics of natural images, it may also prevent such models from capturing contextual long-range feature interactions. In this work, we propose a simple, lightweight approach for better *context exploitation* in CNNs. We do so by introducing a pair of operators: gather, which efficiently aggregates feature responses from a large spatial extent, and excite, which redistributes the pooled information to local features. The operators are cheap, both in terms of number of added parameters and computational complexity, and can be integrated directly in existing architectures to improve their performance. Experiments on several datasets show that gather-excite can bring benefits comparable to increasing the depth of a CNN at a fraction of the cost. For example, we find ResNet-50 with gather-excite operators is able to outperform its 101-layer counterpart on ImageNet with no additional learnable parameters. We also propose a parametric gather-excite operator pair which yields further performance gains, relate it to the recently-introduced *Squeeze-and-Excitation* Networks, and analyse the effects of these changes to the CNN feature activation statistics.

## 1 Introduction

Convolutional neural networks (CNN) [21] are the gold-standard approach to problems such as image classification [20, 35, 9], object detection [32] and image segmentation [3]. Thus, there is a significant interest in improved CNN architectures. In computer vision, an idea that has often improved visual representations is to augment functions that perform local decisions with functions that operate on a larger context, providing a cue for resolving local ambiguities [39]. While the term "context" is overloaded [6], in this work we focus specifically on *feature context*, namely the information captured by the feature extractor responses (i.e. the CNN feature maps) as a whole, spread over the full spatial extent of the input image.

In many standard CNN architectures the receptive fields of many feature extractors are theoretically already large enough to cover the input image in full. However, the *effective* size of such fields is in practice considerably smaller [27]. This may be one factor explaining why improving the use of context in deep networks can lead to better performance, as has been repeatedly demonstrated in object detection and other applications [1, 26, 48].

---

[*]Equal contribution

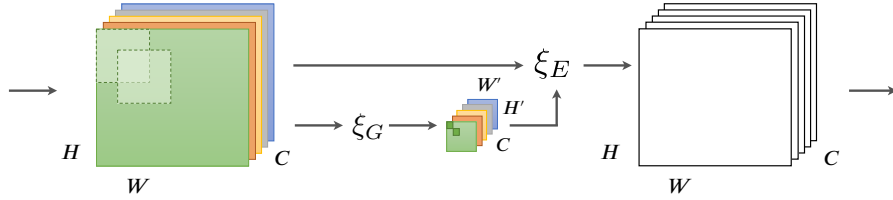

Figure 1: The interaction of a *gather-excite* operator pair, $(\xi_G, \xi_E)$. The gather operator $\xi_G$ first aggregates feature responses across spatial neighbourhoods. The resulting aggregates are then passed, together with the original input tensor, to an excite operator $\xi_E$ that produces an output that matches the dimensions of the input.

Prior work has illustrated that using simple aggregations of low level features can be effective at encoding contextual information for visual tasks, and may prove a useful alternative to iterative methods based on higher level semantic features [44]. Demonstrating the effectiveness of such an approach, the recently proposed Squeeze-and-Excitation (SE) networks [15] showed that reweighting feature channels as a function of features from the full extent of input can improve classification performance. In these models, the *squeeze* operator acts as a lightweight context aggregator and the resulting embeddings are then passed to the reweighting function to ensure that it can exploit information beyond the local receptive fields of each filter.

In this paper, we build on this approach and further explore mechanisms to incorporate context throughout the architecture of a deep network. Our goal is to explore more efficient algorithms as well as the essential properties that make them work well. We formulate these "context" modules as the composition of two operators: a *gather* operator, which aggregates contextual information across large neighbourhoods of each feature map, and an *excite* operator, which modulates the feature maps by conditioning on the aggregates.

Using this decomposition, we chart the space of designs that can exploit feature context in deep networks and explore the effect of different operators independently. Our study leads us to propose a new, lightweight gather-excite pair of operators which yields significant improvements across different architectures, datasets and tasks, with minimal tuning of hyperparameters. We also investigate the effect of the operators on distributed representation learned by existing deep architectures: we find the mechanism produces intermediate representations that exhibit lower *class selectivity*, suggesting that providing access to additional context may enable greater feature re-use. The code for all models used in this work is publicly available at https://github.com/hujie-frank/GENet.

## 2 The Gather-Excite Framework

In this section, we introduce the Gather-Excite (GE) framework and describe its operation.

The design is motivated by examining the flow of information that is typical of CNNs. These models compute a hierarchy of representations that transition gradually from spatial to channel coding. Deeper layers achieve greater abstraction by combining features from previous layers while reducing resolution, increasing the receptive field size of the units, and increasing the number of feature channels.

The family of *bag-of-visual-words* [5, 47, 34] models demonstrated the effectiveness of pooling the information contained in local descriptors to form a global image representation out of a local one. Inspired by this observation, we aim to help convolutional networks exploit the contextual information contained in the field of feature responses computed by the network itself.

To this end, we construct a lightweight function to gather feature responses over large neighbourhoods and use the resulting contextual information to modulate original responses of the neighbourhood elements. Specifically, we define a gather operator $\xi_G$ which aggregates neuron responses over a given spatial extent, and an excite operator $\xi_E$ which takes in both the aggregates and the original input to produce a new tensor with the same dimensions of the original input. The GE operator pair is illustrated in Fig. 1.

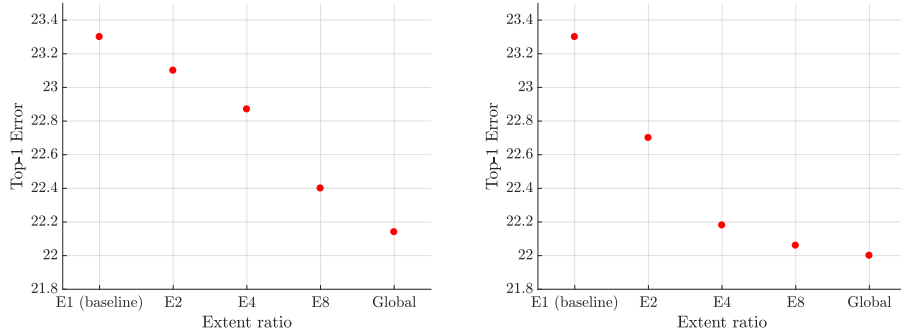

Figure 2: Top-1 ImageNet validation error (%) for the proposed (left) GE-$\theta^-$ and (right) GE-$\theta$ designs based on a ResNet-50 architecture (the *baseline* label indicates the performance of the original ResNet-50 model in both plots). For reference, ResNet-101 achieves a top-1 error of 22.20%. See Sec. 3 for further details.

More formally, let $x = \{x^c : c \in \{1, \ldots, C\}\}$ denote a collection of feature maps produced by the network. To assess the effect of varying the size of the spatial region over which the gathering occurs, we define the selection operator $\iota(u, e) = \{eu + \delta : \delta \in [-\lfloor (2e-1)/2 \rfloor, \lfloor (2e-1)/2 \rfloor]^2\}$ where $e$ represents the *extent ratio* of the selection. We then define a **gather operator** with extent ratio $e$ to be a function $\xi_G : \mathbb{R}^{H \times W \times C} \to \mathbb{R}^{H' \times W' \times C}$ ($H' = \lceil \frac{H}{e} \rceil$, $W' = \lceil \frac{W}{e} \rceil$) that satisfies for any input $x$ the constraint $\xi_G(x)_u^c = \xi_G(x \odot \mathbf{1}_{\iota_{(u,e)}}^c)$, where $u \in \{1, \ldots, H'\} \times \{1, \ldots, W'\}$, $c \in \{1, \ldots, C\}$, $\mathbf{1}_{\{\cdot\}}$ denotes the indicator tensor and $\odot$ is the Hadamard product. This notation simply states that at each output location $u$ of the channel $c$, the gather operator has a receptive field of the input that lies within a single channel and has an area bounded by $(2e-1)^2$. If the field envelops the full input feature map, we say that the gather operator has *global* extent. The objective of the **excite operator** is to make use of the gathered output as a contextual feature and takes the form $\xi_E(x, \hat{x}) = x \odot f(\hat{x})$, where $f : \mathbb{R}^{H' \times W' \times C} \to [0, 1]^{H \times W \times C}$ is the map responsible for rescaling and distributing the signal from the aggregates.

## 3 Models and Experiments

In this section, we explore and evaluate a number of possible instantiations of the gather-excite framework. To compare the utility of each design, we conduct a series of experiments on the task of image classification using the ImageNet 1K dataset [33]. The dataset contains 1.2 million training images and 50k validation images. In the experiments that follow, all models are trained on the training set and evaluated on the validation set. We base our investigation on the popular ResNet-50 architecture which attains good performance on this dataset and has been shown to generalise effectively to a range of other domains [9]. New models are formed by inserting gather-excite operators into the residual branch immediately before summation with the identity branch of each building block of ResNet-50. These models are trained from random initialisation [10] using SGD with momentum 0.9 with minibatches of 256 images, each cropped to $224 \times 224$ pixels. The initial learning rate is set to 0.1 and is reduced by a factor of 10 each time the loss plateaus (three times). Models typically train for approximately 300 epochs in total (note that this produces stronger models than the fixed 100-epoch optimisation schedule used in [15]). In all experiments, we report single-centre-crop results on the ImageNet validation set.

### 3.1 Parameter-free pairings

We first consider a collection of GE pairings which require no additional learnable parameters. We take the gather operator $\xi_G$ to be average pooling with varying extent ratios (the effect of changing the pooling operator is analysed in the suppl. material). The excite operator then resizes the aggregates, applies a sigmoid and multiplies the result with the input. Thus, each output feature map is computed as $y^c = x \odot \sigma(\text{interp}(\xi_G(x)^c))$, where $\text{interp}(\cdot)$ denotes resizing to the original input size via nearest neighbour interpolation. We refer to this model as GE-$\theta^-$, where the notation $\theta^-$ is used to denote

|                        | top-1 err. | top-5 err. | GFLOPs | #Params |
| ---------------------- | ---------- | ---------- | ------ | ------- |
| ResNet-50 (Baseline)   | 23.30      | 6.55       | 3.86   | 25.6 M  |
| GE-$\theta$ (stage2)   | 23.29      | 6.50       | 3.86   | 28.0 M  |
| GE-$\theta$ (stage3)   | 22.70      | 6.24       | 3.86   | 27.2 M  |
| GE-$\theta$ (stage4)   | 22.50      | 6.20       | 3.86   | 26.8 M  |
| GE-$\theta$ (all)      | 22.00      | 5.87       | 3.87   | 31.2 M  |

Table 1: Effect (error %) of inserting GE operators at different stages of the baseline architecture ResNet-50.

that the operator is parameter-free[2]. A diagram illustrating how these operators are integrated into a residual unit can be found in Fig. 4 of the supplementary material.

**Spatial extent:** This basic model allows us to test the central hypothesis of this paper, namely that providing the network with access to simple summaries of additional feature context improves the representational power of the network. To this end, our first experiment varies the spatial extent ratio of the GE-$\theta^-$ design: we consider values of $e = \{2, 4, 8\}$, as well a global extent ratio using global average pooling. The results of this experiment are shown in Fig. 2 (left). Each increase in the extent ratio yields consistent improvements over the performance of the ResNet-50 baseline (23.30% top-1 error), with the global extent ratio achieving the strongest performance (22.14% top-1 error). This experiment suggests that even with a simple parameter-free approach, context-based modulation can strengthen the discriminative power of the network. Remarkably, this model is competitive with the much heavier ResNet-101 model (22.20% top-1 error). In all following experiments, except where noted otherwise, a global extent ratio is used.

## 3.2 Parameterised pairings

We have seen that simple gather-excite operators without learned parameters can offer an effective mechanism for exploiting context. To further explore the design space for these pairings, we next consider the introduction of parameters into the gather function, $\xi_G(\theta)$. In this work, we propose to use strided depth-wise convolution as the gather operator, which applies spatial filters to independent channels of the input. We combine $\xi_G(\theta)$ with the excite operator described in Sec. 3.1 and refer to this pairing as GE-$\theta$.

**Spatial extent:** We begin by repeating the experiment to assess the effect of an increased extent ratio for the parameterised model. For parameter efficiency, varying extent ratios $e$ is achieved by chaining $3 \times 3$ stride 2 depth-wise convolutions ($e/2$ such convolutions are performed in total). For the global extent ratio, a single global depth-wise convolution is used. Fig. 2 (right) shows the results of this experiment. We observe a similar overall trend to the GE-$\theta^-$ study and note that the introduction of additional parameters brings expected improvements over the parameter-free design.

**Effect on different stages:** We next investigate the influence of GE-$\theta$ on different stages (here we use the term "stage" as it is defined in [9]) of the network by training model variants in which the operators are inserted into each stage separately. The accuracy, computational cost and model complexity of the resulting models are shown in Tab. 1. While there is some improvement from insertion at every stage, the greatest improvement comes from the mid and late stages (where there are also more channels). The effects of insertion at different stages are not mutually redundant, in the sense that they can be combined effectively to further bolster performance. For simplicity, we include GE operators throughout the network in all remaining experiments, but we note that if parameter storage is an important concern, GE can be removed from Stage 2 at a marginal cost in performance.

**Relationship to Squeeze-and-Excitation Networks:** The recently proposed Squeeze-and-Excitation Networks [15] can be viewed as a particular GE pairing, in which the gather operator is a parameter-free operation (global average pooling) and the excite operator is a fully connected subnetwork. Given the strong performance of these networks (see [15] for details), a natural question arises: are the benefits of parameterising the gather operator complementary to increasing the capacity of the excite operator? To answer this question, we experiment with a further variant, GE-$\theta^+$,

|  | top-1 err. | top-5 err. | GFLOPs | #Params |
|---|---|---|---|---|
| ResNet-101 | 22.20 | 6.14 | 7.57 | 44.6 M |
| ResNet-50 (Baseline) | 23.30 | 6.55 | 3.86 | 25.6 M |
| SE | 22.12 | 5.99 | 3.87 | 28.1 M |
| GE-$\theta^-$ | 22.14 | 6.24 | 3.86 | 25.6 M |
| GE-$\theta$ | 22.00 | 5.87 | 3.87 | 31.2 M |
| GE-$\theta^+$ | 21.88 | 5.80 | 3.87 | 33.7 M |

Table 2: Comparison of differing GE configurations with a ResNet-50 baseline on the ImageNet validation set (error %) and their respective complexities. The ResNet-101 model is included for reference.

|  | top-1 err. | top-5 err. | GFLOPs | #Params |
|---|---|---|---|---|
| ResNet-152 | 21.87 | 5.78 | 11.28 | 60.3 M |
| ResNet-101 (Baseline) | 22.20 | 6.14 | 7.57 | 44.6 M |
| SE | 20.94 | 5.50 | 7.58 | 49.4 M |
| GE-$\theta^-$ | 21.47 | 5.69 | 7.58 | 44.6 M |
| GE-$\theta$ | 21.46 | 5.45 | 7.59 | 53.7 M |
| GE-$\theta^+$ | **20.74** | **5.29** | 7.59 | 58.4 M |

Table 3: Comparison of differing GE configurations with a ResNet-101 baseline on the ImageNet validation set (error %) and their respective complexities. The GE-$\theta^-$ (101) model outperforms a deeper ResNet-152 (included above for reference).

which combines the GE-$\theta$ design with a $1 \times 1$ convolutional channel subnetwork excite operator (supporting the use of variable spatial extent ratios). The parameterised excite operator thus takes the form $\xi_E(x, \hat{x}) = x \odot \sigma(\text{interp}(f(\hat{x}|\theta)))$, where $f(\hat{x}|\theta)$ matches the definition given in [15], with reduction ratio 16). The performance of the resulting model is given in Tab. 2. We observe that the GE-$\theta^+$ model not only outperforms the SE and GE-$\theta$ models, but approaches the performance of the considerably larger 152 layer ResNet (21.88% vs 21.87% top-1 error) at approximately one third of the computational complexity.

### 3.3 Generalisation

**Deeper networks:** We next ask whether the improvements brought by incorporating GE operators are complementary to the benefits of increased network depth. To address this question, we train deeper ResNet-101 variants of the GE-$\theta^-$, GE-$\theta$ and GE-$\theta^+$ designs. The results are reported in Tab. 3. It is important to note here that the GE operators themselves add layers to the architecture (thus this experiment does not control precisely for network depth). However, they do so in an extremely lightweight manner in comparison to the standard computational blocks that form the network and we observe that the improvements achieved by GE transfer to the deeper ResNet-101 baseline, suggesting that to a reasonable degree, these gains are complementary to increasing the depth of the underlying backbone network.

**Resource constrained architectures:** We have seen that GE operators can strengthen deep residual network architectures. However, these models are largely composed of dense convolutional computational units. Driven by demand for mobile applications, a number of more sparsely connected architectures have recently been proposed with a view to achieving good performance under strict resource constraints [14, 50]. We would therefore like to assess how well GE generalises to such scenarios. To answer this question, we conduct a series of experiments on the ShuffleNet architecture [50], an efficient model that achieves a good tradeoff between accuracy and latency. Results are reported in Tab. 4. In practice, we found these models challenging to optimise and required longer training schedules ($\approx 400$ epochs) to reproduce the performance of the baseline model reported in [50] (training curves under a fixed schedule are provided in the suppl. material). We also found it difficult to achieve improvements without the use of additional parameters. The GE-$\theta$ variants yield improvements in performance at a fairly modest theoretical computational complexity. In scenarios for which parameter storage represents the primary system constraint, a naive application of GE may

| ShuffleNet variant | top-1 err. | top-5 err. | MFLOPs | #Params |
|---|---|---|---|---|
| ShuffleNet (Baseline) | 32.60 | 12.40 | 137.5 | 1.9 M |
| SE | 31.24 | 11.38 | 139.9 | 2.5 M |
| GE-$\theta$ (E2) | 32.40 | 12.31 | 138.9 | 2.0 M |
| GE-$\theta$ (E4) | 32.32 | 12.24 | 139.1 | 2.1 M |
| GE-$\theta$ (E8) | 32.12 | 12.11 | 139.2 | 2.2 M |
| GE-$\theta$ | 31.80 | 11.98 | 140.8 | 3.6 M |
| GE-$\theta^+$ | **30.12** | **10.70** | 141.6 | 4.4 M |

Table 4: Comparison of differing GE configurations with a ShuffleNet baseline on the ImageNet validation set (error %) and their respective complexities. Here, ShuffleNet refers to "ShuffleNet $1 \times (g = 3)$" in [50].

| | ResNet-110 [10] | ResNet-164 [10] | WRN-16-8 [49] |
|---|---|---|---|
| Baseline | 6.37 / 26.88 | 5.46 / 24.33 | 4.27 / 20.43 |
| SE | 5.21 / 23.85 | 4.39 / 21.31 | 3.88 / 19.14 |
| GE-$\theta^-$ | 6.01 / 26.58 | 5.12 / 23.94 | 4.12 / 20.25 |
| GE-$\theta$ | 5.57 / 24.29 | 4.67 / 21.86 | 4.02 / 19.76 |
| GE-$\theta^+$ | **4.93 / 23.36** | **4.07 / 20.85** | **3.72 / 18.87** |

Table 5: Classification error (%) on the CIFAR-10/100 test set with standard data augmentation (padding 4 pixels on each side, random crop and flip).

be less appropriate and more care is needed to achieve a good tradeoff between accuracy and storage (this may be achieved, for example, by using GE at a subset of the layers).

**Beyond ImageNet:** We next assess the ability of GE operators generalise to other datasets beyond ImageNet. To this end, we conduct additional experiments on the CIFAR-10 and CIFAR-100 image classification benchmarks [19]. These datasets consist of $32 \times 32$ color images drawn from 10 classes and 100 classes respectively. Each contains 50k train images and 10k test images. We adopt a standard data augmentation scheme (as used in [9, 16, 24]) to facilitate a useful comparative analysis between models. During training, images are first zero-padded on each side with four pixels, then a random $32 \times 32$ patch is produced from the padded image or its horizontal flip before applying mean/std normalization. We combine GE operators with several popular backbones for CIFAR: ResNet-110 [10], ResNet-164 [10] and the Wide Residual Network-16-8 [49]. The results are reported in Tab. 5. We observe that even on datasets with considerably different characteristics (e.g. $32 \times 32$ pixels), GE still yields good performance gains.

**Beyond image classification:** We would like to evaluate whether GE operators can generalise to other tasks beyond image classification. For this purpose, we train an object detector on MS COCO [25], a dataset which has approximately 80k training images and 40k validation images (we use the train-val splits provided in the 2014 release). Our experiment uses the Faster R-CNN framework [32] (replacing the *RoIPool* operation with *RoIAlign* proposed in [8]) and otherwise follows the training settings in [9]. We train two variants: one with a ResNet-50 backbone and one with a GE-$\theta$ (E8) backbone, keeping all other settings fixed. The ResNet-50 baseline performance is 27.3% mAP. Incorporating the GE-$\theta$ backbone improves the baseline performance to 28.6% mAP.

## 4 Analysis and Discussion

**Effect on learned representations:** We have seen that GE operators can improve the performance of a deep network for visual tasks and would like to gain some insight into how the learned features may differ from those found in the baseline ResNet-50 model. For this purpose, we use the *class selectivity index* metric introduced by [28] to analyse the features of these models. This metric computes, for each feature map, the difference between the highest class-conditional mean activity and the mean of all remaining class-conditional activities over a given data distribution. The resulting measurement is normalised such that it varies between zero and one, where one indicates that a filter only fires for a single class and zero indicates that the filter produced the same value for every class. The metric is of interest to our work because it provides some measure of the degree to which features are being

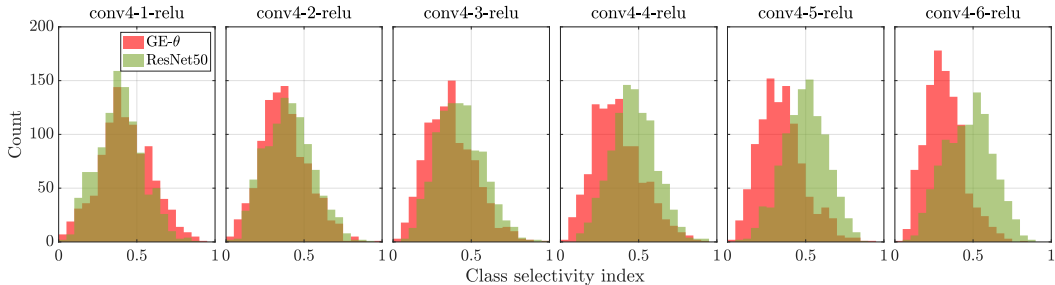

Figure 3: Each figure depicts the class selectivity index distribution for features in both the baseline ResNet-50 and corresponding GE-$\theta$ network at various blocks in the fourth stage of their architectures. As depth increases, we observe that the GE-$\theta$ model exhibits less class selectivity than the ResNet-50 baseline.

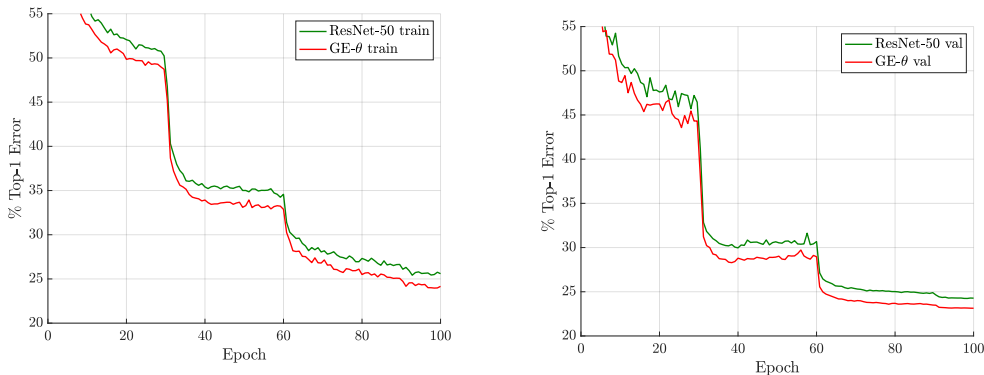

Figure 4: Top-1 error (%) on the ImageNet training set (left) and validation set (right) of the ResNet-50 baseline and proposed GE-$\theta$ (global extent) model under a fixed-length training schedule.

shared *across classes*, a central property of distributed representations that can describe concepts efficiently [12].

We compute the class selectivity index for intermediate representations generated in the fourth stage (here we use the term "stage" as it is defined in [9]). The features of this stage have been shown to generalise well to other semantic tasks [31]. We compute class selectivity histograms for the last layer in each block in this stage of both models, and present the results of GE-$\theta$ and ResNet-50 in Fig. 3. An interesting trend emerges: in the early blocks of the stage, the distribution of class selectivity for both models appears to be closely matched. However, with increasing depth, the distributions begin to separate, and by `conv4-6-relu` the distributions appear more distinct with GE-$\theta$ exhibiting less class selectivity than ResNet-50. Assuming that additional context may allow the network to better recognise patterns that would be locally ambiguous, we hypothesise that networks without access to such context are required to allocate a greater number of highly specialised units that are devoted to the resolution of these ambiguities, reducing feature re-use. Additional analyses of the SE and GE-$\theta^-$ models can be found in the suppl. material.

**Effect on convergence:** We explore how the usage of GE operators play a role in the optimisation of deep networks. For this experiment, we train both a baseline ResNet-50 and a GE-$\theta$ model (with global extent ratio) from scratch on ImageNet using a fixed 100 epoch schedule. The learning rate is initialised to 0.1 and decreased by a factor of 10 every 30 epochs. The results of this experiment are shown in Fig. 4. We observe that the GE-$\theta$ model achieves lower training and validation error throughout the course of the optimisation schedule. A similar trend was reported when training with SE blocks [15], which as noted in Sec. 3.2, can be interpreted as a parameter-free gather operator and a parameterised excite operator. By contrast, we found empirically that the GE-$\theta^-$ model does not exhibit the same ease of optimisation and takes longer to learn effective representations.

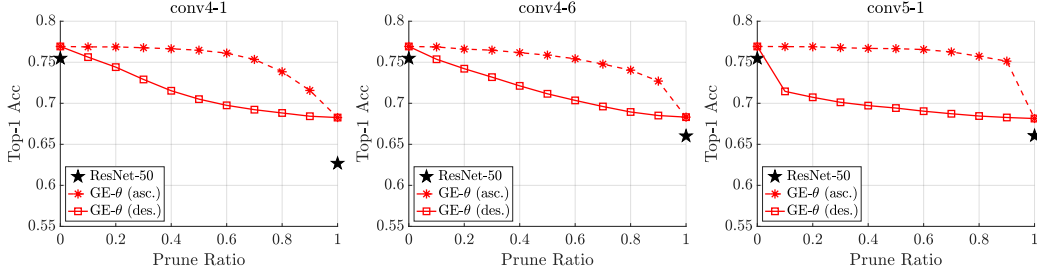

Figure 5: Top-1 ImageNet validation accuracy for the GE-$\theta$ model after dropping a given ratio of feature maps out the residual branch for each test image. Dashed line denotes the effect of dropping features with the least assigned importance scores first. Solid line denotes the effect of dropping features with the highest assigned importance scores first. For reference, the black stars indicate the importance of these feature blocks to the ResNet-50 model (see Sec. 4 for further details).

**Feature importance and performance.** The gating mechanism of the excite operator allows the network to perform feature selection throughout the learning process, using the feature importance scores that are assigned to the outputs of the gather operator. Features that are assigned a larger importance will be preserved, and those with lower importance will be squashed towards zero. While intuitively we might expect that feature importance is a good predictor of the contribution of a feature to the overall network performance, we would like to verify this relationship. We conduct experiments on a GE-$\theta$ network, based on the ResNet-50 architecture. We first examine the effect of pruning the least important features: given a building block of the models, for each test image we sort the channel importances induced by the gating mechanism in ascending order (labelled as "asc." in Fig. 5), and set a portion (the prune ratio) of the values to zero in a first-to-last manner. As the prune ratio increases, information flow flows through an increasingly small subset of features. Thus, no feature maps are dropped out when the prune ratio is equal to zero, and the whole residual branch is dropped out when the ratio is equal to one (i.e., the information of the identity branch passes through directly). We repeat this experiment in reverse order, dropping out the most important features first (this process is labelled "des." in Fig. 5). This experiment is repeated for three building blocks in GE-$\theta$ (experiments for SE are included in the suppl. material). As a reference for the relative importance of features contained in these residual branches, we additionally report the performance of the baseline ResNet-50 model with the prune ratio set to 0 and 1 respectively. We observe that preserving the features estimated as most important by the excite operator retains the much of the overall accuracy during the early part of the pruning process before an increasingly strong decay in performance occurs. When reversing the pruning order, the shape of this performance curve is inverted, suggesting a consistent positive correlation between the estimated feature importance and overall performance. This trend is clearest for the deeper `conv5-1` block, indicating a stronger dependence between primary features and concepts, which is consistent with findings in previous work [22, 28]. While these feature importance estimates are instance-specific, they can also be used to probe the relationships between classes and different features [15], and may potentially be useful as a tool for interpreting the activations of networks.

## 5 Related Work

Context-based features have a rich history of use in computer vision, motivated by studies in perception that have shown that contextual information influences the accuracy and efficiency of object recognition and detection by humans [2, 13]. Several pioneering automated vision systems incorporated context as a component of sophisticated rule-based approaches to image understanding [36, 7]; for tasks such as object recognition and detection, low-dimensional, global descriptors have often proven effective as contextual clues [39, 30, 40]. Alternative approaches based on graphical models represent another viable mechanism for exploiting context [11, 29] and many other forms of contextual features have been proposed [6]. A number of works have incorporated context for improving semantic segmentation (e.g. [48, 23]), and in particular, ParseNet [26] showed that encoding context through global feature averaging can be highly effective for this task.

The Inception family of architectures [38, 37] popularised the use of multi-scale convolutional modules, which help ensure the efficient aggregation of context throughout the hierarchy of learned representations [17]. Variants of these modules have emerged in recent work on automated architecture search [51], suggesting that they are components of (at least) a local optimum in the current design space of network blocks. Recent work has developed both powerful and generic parameterised attention modules to allow the system to extract informative signals dynamically [46, 4, 45]. Top-down attention modules [42] and self-attention [41] can be used to exploit global relationships between features. By reweighting features as a generic function of all pairwise interactions, *non-local networks* [43] showed that self-attention can be generalised to a broad family of global operator blocks useful for visual tasks.

There has also been considerable recent interest in developing more specialised, lightweight modules that can be cheaply integrated into existing designs. Our work builds on the ideas developed in *Squeeze-and-Excitation* networks [15], which used global embeddings as part of the SE block design to provide context to the recalibration function. We draw particular inspiration from the studies conducted in [44], which showed that useful contextual information for localising objects can be inferred in a feed-forward manner from simple summaries of basic image descriptors (our aim is to incorporate such summaries of low, mid and high level features throughout the model). In particular, we take the SE emphasis on lightweight contextual mechanisms to its logical extreme, showing that strong performance gains can be achieved by the GE-$\theta^-$ variant with no additional learnable parameters. We note that similar parameterised computational mechanisms have also been explored in the image restoration community [18], providing an interesting alternative interpretation of this family of module designs as learnable activation functions.

## 6 Conclusion and Future Work

In this work we considered the question of how to efficiently exploit feature context in CNNs. We proposed the gather-excite (GE) framework to address this issue and provided experimental evidence that demonstrates the effectiveness of this approach across multiple datasets and model architectures. In future work we plan to investigate whether gather-excite operators may prove useful in other computer vision tasks such as semantic segmentation, which we anticipate may also benefit from efficient use of feature context.

**Acknowledgments.** The authors would like to thank Andrew Zisserman and Aravindh Mahendran for many helpful discussions. Samuel Albanie is supported by ESPRC AIMS CDT. Andrea Vedaldi is supported by ERC 638009-IDIU.

## Footnotes

[2]Throughout this work, we use the term "parameter-free" to denote a model that requires no additional learnable parameters. Under this definition, average pooling and nearest neighbour interpolation are parameter-free operations.

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
