[Supplementary Material]

# Gather-Excite: Exploiting Feature Context in Convolutional Neural Networks
# Supplementary Material

**Optimization curves.** In Fig. 1 we show the effect of training ShuffleNet and the GE-$\theta$ variant under a fixed, 100 epoch schedule to allow a direct comparison. As noted in main paper, for the results reported in Sec. 3 we used a longer training schedule ($\approx 400$ epochs) to reproduce the baseline ShuffleNet performance.

Figure 1: Top-1 error (%) on (left) the ImageNet training set and (right) the ImageNet validation set of the ShuffleNet Baseline and GE-$\theta$ variant of this architecture (with global extent) trained with a fixed 100 epoch schedule.

| GE-$\theta^-$ variant | top-1 err. | top-5 err. |
|---|---|---|
| ResNet-50 | 23.30 | 6.55 |
| (E4, max) | 23.10 | 6.39 |
| (E4, avg) | 22.87 | 6.40 |
| (global, max) | 23.65 | 6.62 |
| (global, avg) | **22.14** | **6.24** |

Table 1: Influence (error %) of different pooling operators for parameter-free GE designs. Each GE-$\theta^-$ variant is described by its (`extent-ratio, pooling-type`) pair.

**Pooling method.** We conduct additional experiments to assess the influence of the pooling method for GE-$\theta^-$ networks (shown in Tab. 1). Average pooling aggregates the neighbouring elements with equal contribution, while max pooling picks a single element to represent its neighbours. We observe that average pooling consistently outperforms max pooling. While adding contextual information by max pooling over a moderate extent can help the baseline model, it hurts the performance when the full global extent is used. This suggests that in contrast to averaging, a naive, parameter-free application of max pooling makes poor use of contextual information and inhibits, rather than facilitates, its

efficient propagation. However, we note that when additional learnable parameters are introduced, this may no longer be the case: an interesting study presented in [1] has shown that there can be benefits to combining the output of both max and average pooling.

**Class selectivity indices.** Following the approach described in Sec. 4 of the paper, we compute histograms of the class selectivity indices for GE-$\theta^-$ and SE models and compare them with the baseline ResNet-50 in Fig. 2. We observe a weaker, but similar trend to the histograms of GE-$\theta$ reported in the main paper, characterised by a gradually emerging gap between the histograms of features at greater depth in stage four.

Figure 2: Each figure compares the class selectivity index distribution of the features of ResNet-50 against the GE-$\theta^-$ (top row) and SE (bottom row) networks at various blocks in the fourth stage of their architectures.

**Feature importance.** We repeat the pruning experiments described in Sec. 4 of the main paper (under the paragraph entitled *Feature importance and performance*) for an SE network based on a ResNet-50 backbone. The results of this experiment are reported in Fig. 3. We observe that the curves broadly match the trends seen in the GE-$\theta$ curves depicted in the main paper.

Figure 3: Top-1 ImageNet validation accuracy for the SE model after dropping a ratio of feature maps out for each test image. Dashed lines denote the effect of dropping features with the least assigned importance scores first. Solid lines denote the effect of dropping features with the highest assigned importance scores first. For reference, the black stars indicate the importance of these feature blocks to the ResNet-50 model.

**Operator diagrams.** In Fig. 4, we illustrate diagrams of several of the GE variants described in the main paper.

Figure 4: The schema of Gather-Excite modules. **Top-left:** GE-$\theta^-$ (E8). **Top-right:** GE-$\theta^-$. **Bottom-left:** GE-$\theta$ (E8). **Bottom-right:** GE-$\theta$.

# References

[1] Sanghyun Woo, Jongchan Park, Joon-Young Lee, and In So Kweon. CBAM: Convolutional block attention module. In *ECCV*, 2018. 2