[Reviews · NeurIPS 2018]

Reviewer 1



This paper proposes a new component for residual networks to improve the performance on image classification task. The proposed idea is to find channel-specific attention from each feature maps, which is to gather the spatial information by a particular layer such as the average pooling, and then the scatter layer such as nearest neighbor upsampling is followed. Experiments are done on the datasets including ImageNet-1k and CIFAR-datasets, and the study of the class sensitivity compared with the original ResNets is provided to support the effectiveness of the proposed method. Pros) +) The proposed spatial attention idea is simple yet promising. +) The authors provide sufficient experimental grounds. +) The class sensitivity study could be a strong supportive ground for the proposed method. Cons) +) Some notations are not clearly presented throughout the paper. +) There is no detailed architecture description of GS-P. +) Eventually, the global average pooling is the best. Then, the spatial information may not be useful, and the proposed method is quite similar to SE-Net. +) The authors claim that the method does not involve extra parameters, but they mainly do experiments with GS-PP which uses depth-wise convolution layers that clearly have additional parameters. The authors should refine the writing. Further comments) 1) Please explain GS-P architecture in detail. Specifically, please explain the scatter operation using depth-wise convolutions in detail. 2) With GS-P, How did you achieve several extent ratios? How did you get the global extent ratio with it? 3) Please explain why GS-PP-50 (seems to use depth-wise convs + fcs) could outperform ResNet-SE-50? 4) It is better to show the result of GS-PP-101 and ResNet-101-SE in Table 3. 5) It is better to show the result of GS-P-SE in Table 2, 3 and 4. 6) How about applying GS-net on the second or third stage features? 7) Could you provide any intuitions why the global average pooling is better than the operator with a smaller extent ratio?

Reviewer 2



This paper present a module in neural networks called Gather-Scatter (GS), which is used to provide context propagation in representation learning. The GS module is very simple, in this paper, average pooling and additional convolutional layers are used to coonstruct the GS module. The draft is clearly written and well organized. The paper is very clear and the author provided enough training detailes to reproduce the results. Besides, a series of baseline models and datasets are used to show the effectiveness of proposed module, including ResNet-50, ResNet-101, ShuffleNet, ImageNet, CIFAR-10 and CIFAR-100. Useful ablation studies are presented to help understand the proposed method. However, the main drawback of this paper is on the novelty. There are strong similarities to recently proposed Squeeze-and-Excitation Networks, and there is no comparison to this work. Besides, the some analysis is not convincing enough, and it still needs more studies to lead a stronger paper. As a conclusion, I rate this paper as "Marginally below the acceptance threshold". In the following, the detailed weakness of this paper are listed: 1) The proposed method is very similar to Squeeze-and-Excitation Networks [1], but there is no comparison to the related work quantitatively. 2) There is only the results on image classification task. However, one of success for deep learning is that it allows people leverage pretrained representation. To show the effectiveness of this approach that learns better representation, more tasks are needed, such as semantic segmentation. Especially, the key idea of this method is on the context propagation, and context information plays an important role in semantic segmentation, and thus it is important to know. 3) GS module is used to propagate the context information over different spatial locations. Is the effective receptive field improved, which can be computed from [2]? It is interesting to know how the effective receptive field changed after applying GS module. 4) The analysis from line 128 to 149 is not convincing enough. From the histogram as shown in Fig 3, the GS-P-50 model has smaller class selectivity score, which means GS-P-50 shares more features and ResNet-50 learns more class specific features. And authors hypothesize that additional context may allow the network to reduce its dependency. What is the reason such an observation can indicate GS-P-50 learns better representation? Reference: [1] J. Hu, L. Shen and G. Sun, Squeeze-and-Excitation Networks, CVPR, 2018. [2] W. Luo et al., Understanding the Effective Receptive Field in Deep Convolutional Neural Networks, NIPS, 2016.

Reviewer 3



This paper proposes to augment standard deep residual networks for image classification with explicit gather-scatter operations, in order to improve propagation of useful contextual information. It implements a per-channel gather-scatter operation. The gather phase performs either global average pooling or max pooling (termed "picking") per channel, over an extended spatial domain. The scatter phase upsamples the gathered feature maps back to the original size, and combines them with the input feature maps. Experiments show integrating gather-scatter operations into residual networks yields a performance boost on the standard ImageNet classification task without any, or with minimal, parameter increase. The paper justifiably discusses connections with Squeeze-and-Excitation networks [10]. However, it is missing citation, discussion, and comparison to two other highly related recent publications: (1) Multigrid Neural Architectures Tsung-Wei Ke, Michael Maire, Stella X. Yu CVPR 2017 (2) Non-local Neural Networks Xiaolong Wang, Ross Girshick, Abhinav Gupta, Kaiming He CVPR 2018 The first of these works, [Ke et al, CVPR 2017] extends residual networks with a scale-space information propagation mechanism: at each layer, information flows both up and down a scale-space pyramid. This is implemented by combining convolution operations with max-pooling, upsampling, and concatenation at every layer in the network. These ingredients sound quite similar to the proposed gather-scatter scheme and, in fact, could be viewed as a generalization of the proposed gather-scatter approach to one which operates at all spatial scales. Like the current paper, [Ke et al] specifically mention rapid integration of contextual cues and demonstrate improved accuracy and parameter savings on ImageNet classification using their multigrid residual network. The second of these related works, [Wang et al, CVPR 2018], suggests inserting a non-local information gathering operation into neural networks, similar in formulation to the gather operation proposed here. They discuss multiple functional forms and also present results of experiments modifying residual networks to include their proposed operations. The existence of both of these published prior works decreases the relative novelty of the approach proposed here. The fact that neither are cited is a serious problem. Ideally, both would be cited, discussed, and compared against in experiments. If the particular implementation proposed by this paper has advantages, a thorough experimental comparison to prior work is needed to justify them. In light of this prior work, I do not believe the idea or current results alone are sufficiently novel or impressive to carry the paper. --- The rebuttal addresses my concerns about discussion of related work and addition of experimental comparison to that work. I have raised my overall score.